# Brain Stimulation in Eating Disorders: State of the Art and Future Perspectives

**DOI:** 10.3390/jcm9082358

**Published:** 2020-07-23

**Authors:** Philibert Duriez, Rami Bou Khalil, Yara Chamoun, Redwan Maatoug, Robertas Strumila, Maude Seneque, Philip Gorwood, Philippe Courtet, Sébastien Guillaume

**Affiliations:** 1GHU Paris Psychiatry and Neuroscience, Clinique des Maladies Mentales et de l’Encéphale (CMME), Sainte-Anne Hospital, 75014 Paris, France; phduriez@gmail.com (P.D.); P.GORWOOD@ghu-paris.fr (P.G.); 2Institute of Psychiatry and Neurosciences of Paris (IPNP), UMR_S1266, INSERM, Université de Paris, 102-108 rue de la Santé, 75014 Paris, France; 3Department of Psychiatry, Hotel Dieu de France- Saint Joseph University, 166830 Beirut, Lebanon; ramiboukhalil@hotmail.com (R.B.K.); yara.chamoun@hotmail.com (Y.C.); 4Neuropsychiatry: Epidemiological and Clinical Research, Université Montpellier, INSERM, CHU de Montpellier, 34295 Montpellier, France; maude.seneque@gmail.com (M.S.); p-courtet@chu-montpellier.fr (P.C.); 5Sorbonne Université, AP-HP, Service de Psychiatrie Adulte de la Pitié-Salpêtrière, Institut du Cerveau, ICM, 75013 Paris, France; redwanmaatoug@gmail.com; 6Faculty of Medicine, Institute of Clinical Medicine, Psychiatric Clinic, Vilnius University, 03101 Vilnius, Lithuania; robertas.strumila@gmail.com; 7Department of Emergency Psychiatry and Post-Acute Care, CHRU Montpellier, 34295 Montpellier, France

**Keywords:** rTMS, deep brain stimulation, treatment, anorexia, bulimia, binge eating disorders

## Abstract

The management of eating disorders (EDs) is still difficult and few treatments are effective. Recently, several studies have described the important contribution of non-invasive brain stimulation (repetitive transcranial magnetic stimulation, transcranial direct current stimulation, and electroconvulsive therapy) and invasive brain stimulation (deep brain stimulation and vagal nerve stimulation) for ED management. This review summarizes the available evidence supporting the use of brain stimulation in ED. All published studies on brain stimulation in ED as well as ongoing trials registered at clinicaltrials.gov were examined. Articles on neuromodulation research and perspective articles were also included. This analysis indicates that brain stimulation in EDs is still in its infancy. Literature data consist mainly of case reports, cases series, open studies, and only a few randomized controlled trials. Consequently, the evidence supporting the use of brain stimulation in EDs remains weak. Finally, this review discusses future directions in this research domain (e.g., sites of modulation, how to enhance neuromodulation efficacy, personalized protocols).

## 1. Introduction

Eating disorders (EDs) are serious psychiatric disorders characterized by abnormal eating or weight-control behaviors [1]. These disorders are most often chronic and relapsing, and this has a heavy impact on the patients’ physical and mental health and life expectancy. Anorexia nervosa (AN) is a multifactorial ED characterized by significantly low body weight for the individual’s height, age, and developmental stage, intense fear of gaining weight despite obvious thinness, and extreme behaviors designed to lose weight, such as food restriction with or without induced vomiting, or use of laxatives. The consequence is massive weight loss and/or pathological thinness. Binge eating disorder (BED) and bulimia nervosa (BN) are EDs characterized by recurrent episodes of binge eating and loss of personal control during binging. Individuals with BN counteract binge eating with compensatory behaviors to prevent weight gain, whereas individuals with BED do not exhibit recurrent compensatory behaviors. BED and BN are characterized by compulsive overeating, and shared neural alterations and neurobiological mechanisms underlie these EDs. EDs impair the quality of life of both patients and their families [2]. Among individuals with EDs, mortality and morbidity are increased and health service use is particularly high, resulting in elevated healthcare costs [3].

To date, ED management is still difficult, and few treatments have demonstrated their efficacy. In accordance with most guidelines, management programs are generally multidisciplinary. In AN, the available treatment recommendations aim at restoring normal weight, adapting and relaxing eating behaviors, improving social and interpersonal relationships, and improving the patient’s self-image. Psychotherapy is the main therapeutic approach. However, data to guide the psychotherapy choice are limited and controversial [4]. Medication trials have been disappointing [5]. Overall, about one-third of patients will not recover, and the standardized mortality rate is 5.9 [6]. In patients with BN or BED, the best validated and most frequently used treatment is ED-specific cognitive behavioral therapy [7]. Nevertheless, a meta-analysis found that core BN symptoms are still present in more than 60% of patients, even after receiving the best available treatments [8]. Other psychotherapies have been proposed. Moreover, serotonergic antidepressants, the most frequently used pharmacological option, improve the medium-term symptomatology to some degree, but do not allow full remission. In this context, the development of alternative therapeutic strategies is crucial.

Brain stimulation is a therapeutic modality in which the activity of a specific neural circuit is modified by applying an electric current with predefined frequency, amplitude, and pulse width to restore the functional state without any tissue damage [9]. Brain stimulation is obtained by invasive and non-invasive interventions that stimulate or block the action potentials in the nervous system [9,10]. The ultimate aim of brain stimulation is to reverse maladaptive neurocircuitry changes in neural tissue and improve inter-neuronal connectivity [9,10,11]. These changes implicate synaptic potentiation or depression via regulation of neurotransmitters and ion channels and modification of the expression of intracellular second messengers [12,13]. In direct electrical stimulation, electrodes are used to apply a potential gradient across a neuron that induces intracellular ionic current flow, localized depolarization and hyperpolarization of the cell membrane, resulting in neural stimulation or inhibition [9]. The mechanism for magnetic stimulation is similar, except that the potential gradients are induced in the tissue by a rapidly changing strong magnetic field that is implemented transcutaneously [9]. Non-invasive brain stimulation (NIBS) techniques include electroconvulsive therapy (ECT), repetitive transcranial magnetic stimulation (rTMS), and transcranial direct current stimulation (tDCS). In ECT, seizures are induced by a direct current passing through the brain under general anesthesia [14]. In rTMS, excitability is induced by delivering magnetic stimulation pulses with a wire coil placed externally on the scalp above specific cortical regions [15]. In tDCS, direct current is delivered through electrodes implanted on the scalp, without seizure induction, with the aim of exciting (anodal stimulation) or inhibiting (cathodal stimulation) the activity in specific brain regions [16,17]. Invasive neuromodulation includes deep brain stimulation (DBS) and vagal nerve stimulation (VNS). In DBS, a pulse generator usually implanted in the subclavicular region delivers electrical current to the brain parenchyma through implanted electrodes. On the other hand, in VNS the left cervical vagal nerve is stimulated through a device implanted in the left chest wall [18].

The number of studies on brain stimulation in EDs and/or in dimensions associated with EDs is progressively increasing. In this review, we present the available data, issues, and future directions concerning the use of brain stimulation in EDs.

## 2. Methods

This is a non-systematic review of the literature on brain stimulation techniques used in patients with EDs. Nevertheless, a systematic research of the available literature was performed in the PubMed database by combining the search terms “eating disorders”, “anorexia nervosa”, “bulimia nervosa”, “binge eating disorders” with the terms “neuromodulation”, “rTMS”, “tDCS”, “DBS”, “ECT”, “VNS”. All papers in English or French published up to June 2020 were retrieved. All original articles on brain stimulation in patients with EDs were included. Studies in high risk populations without EDs (e.g., patients with obesity or healthy subjects with high levels of food craving) were not included. This search was then supplemented by internet searches and manual search of the reference lists of potentially relevant articles and reviews, and by examining pertinent trials registered at clinicaltrial.gov. Among the 121 manuscripts identified, after duplicate removal, 72 were screened. All manuscripts are not cited because when a recent systematic review was available, it was included and prioritized in the synthesis of results. Additional studies on neuromodulation mechanisms, ED treatment, potential ED biomarkers, and mechanistic models of circuitries involved in EDs were also added. The search was performed independently by three researchers (P.D., R.B.K., Y.C.). Disagreements were resolved by consensus among all authors.

## 3. Results

### 3.1. rTMS in Eating Disorders

#### 3.1.1. Anorexia Nervosa

As shown in Table 1, In 2008, the first published case on rTMS concerned a patient with AN and comorbid depression. The study reported weight gain and amelioration of the depressive symptoms after 41 sessions of rTMS over the left dorsolateral prefrontal cortex (DLPFC) [19]. This was followed by other case reports and case series on left DLPFC rTMS in patients with AN, often with comorbid depression [20,21,22]. The first case series on five patients with AN and without a clinically manifest comorbidity reported AN symptom improvement at 6 months but no longer at 12 months after 20 sessions of rTMS delivered over the left DLPFC [23]. A pilot study in which one session of high-frequency rTMS (10 Hz) was delivered to the left DLPFC in 10 patients showed that the procedure was well tolerated, leading to a reduction in the sensation of feeling full, fat, and anxious as well as in the urge to exercise after exposure to visual and real food stimuli [24]. These studies were followed by double-blind, controlled clinical trials [25,26,27]. In one of these trials, one rTMS session was delivered to the left DLPFC in 49 patients with AN, but the effect on AN core symptoms was not significant [25] compared with the sham session group. However, in the group analysis, AN core symptoms improved in the rTMS arm compared with baseline (before rTMS), and the results persisted at 24 h of follow-up [25]. Several studies suggest that a higher number of rTMS sessions give better clinical results [26,27]. An ancillary analysis of the sample of a randomized controlled feasibility trial on rTMS [26], using a food choice task, found a decrease in self-controlled food choices, suggesting than rTMS may promote more flexibility in relation to food choices [28]. Finally, the feasibility and safety of rTMS of DLPFC in patients with AN have been confirmed by these randomized controlled trials [25,26,27].

Besides DLPFC, one study on rTMS of the inferior parietal cortex was stopped for safety reasons (NCT01717079). The study promoter informed us that the study was halted following two suicide attempts after the first inclusions. Another pilot case series (*n* = 8 patients with long-lasting AN) used deep rTMS to target the insula and found that this approach was safe and well-tolerated. At the end of the 42 sessions, AN-related obsessions and compulsions were reduced, as were the depression and anxiety scores [30].

#### 3.1.2. Bulimia Nervosa and Binge-Eating Disorder

In a recent review of the literature, Dalton et al. identified eight studies on rTMS for BN. The areas stimulated were mainly the left DLPFC, and the dorsomedial prefrontal cortex (DMPFC). The results showed a decrease in food craving, and in some studies a reduction in food binging or purging behavior (for review see Dalton et al. [44]). DLPFC stimulation may also improve the inhibitory control and decision making in patients with BN [43]. A study on the correlation between salivary cortisol and rTMS on the left DLPFC in BN showed that salivary cortisol concentrations were significantly lower in the rTMS arm compared with the sham rTMS arm. This suggests that stimulation of this area modifies the hypothalamic–pituitary–adrenal axis activity in people with BN [36]. Moreover, frontal lobe oxygenation is decreased after one rTMS session [41]. On the other hand, one randomized controlled trial showed no improvement of binge/purge behavior after 10 sessions of high-frequency rTMS on the left DLPFC [42]. A groundbreaking research study demonstrated that self-reported food craving during exposure to experimental foods remained stable before and after stimulation of the left DLPFC compared with the sham group in which craving increased [45].

Another interesting finding is the relationship between frontostriatal connectivity and response to 20 sessions of rTMS in patients with refractory binge/purge behavior. In this case series, enhanced frontostriatal connectivity was associated with binge/purge behavior improvement after DMPFC-rTMS. Conversely, in non-responders, who showed high connectivity on the resting-state fMRI, rTMS caused paradoxical suppression of frontostriatal connectivity [40]. Finally, in an open-label case series that included patients with various EDs (mainly binge form) and comorbid post-traumatic stress disorder, post-traumatic stress symptoms were improved after 20 to 30 sessions of high-frequency rTMS on the DMPFC [29].

Hence, rTMS is an important tool to explore the neurobiology of craving and binge eating (BE). An ongoing study is investigating rTMS tolerability and safety and its effects on food craving and BE behavior, and also on ED-related psychopathology (including depression, anxiety, and stress symptoms), anthropometric measures, cognition, brain structure and function, hormones, and inflammatory biomarkers [46].

#### 3.1.3. rTMS and Site of Modulation

As shown in Table 1, most of the published studies targeted the left DLPFC with excitatory modulation. Nevertheless, other neuroanatomical targets should be investigated. The first is the DMPFC, which plays an important role in self-control, including self-inhibition of movements, self-cessation of loss-chasing in pathological gamblers, self-suppression of emotional responses, and impulse control. It has been hypothesized that DMPFC stimulation may alter the DMPFC top-down executive control of the striatal regions associated with the urge to binge, thereby improving binge symptoms. The available open label studies using 10 Hz stimulation [40] emphasize the interest of this target particularly in BN and BED, although more work is needed to allow definitive conclusions.

The insula has also been considered in one pilot study. The insula is involved in the process of imagination of food images and food intake, in the perception of taste during food intake and of satiety, and in the feeling of disgust after eating. The insular lobe plays a role in the disturbed body image perception in patients with AN and is implicated in the brain response to stress through its close connection with the hypothalamus–pituitary–adrenal axis [12]. This brain area is reduced in patients with AN [47]. Moreover, Kaye et al. demonstrated a functional disconnection between the insula and ventral caudal putamen in patients with AN in a hungry state [48]. All these findings suggest that insula is a potential target in AN.

The corticostriatal circuits through the orbitofrontal cortex (OFC) might also represent a valuable target. They play key roles in complex human behaviors, such as evaluation, affect regulation, and reward-based decision making, and have been implicated in all ED types [49]. A functional hyperconnectivity between NAcc and the orbitofrontal cortex has been shown in AN [50]. Few studies have targeted this area with NIBS, and to our knowledge, it has not been done in patients with ED. Nevertheless, the good response to inhibitory (1 Hz) rTMS in patients with obsessive-compulsive disorder (OCD) who share some features with ED (e.g., functional hyperconnectivity between NAcc and OFC) [51] supports testing this target in ED.

Finally, other targets have also been explored in patients with addictions with promising results, particularly high-frequency stimulation of the superior frontal gyrus [52] and inhibition of the medial prefrontal cortex in patients with cocaine use disorder [53]. All these areas might constitute interesting targets in bulimic disorders, but it is challenging to decide which region to stimulate to obtain the best results.

### 3.2. tDCS and Eating Disorders

As shown in Table 2, in AN, tDCS approaches on the right and left DLPFC have been tested [54]. In an open-label single-arm study, seven patients with AN underwent left DLPFC anodal tDCS for 25 min daily for 10 days. In three patients, the levels of eating and depressive symptoms improved immediately after the sessions, and the response was maintained at 1 month in one patient [55]. Of note, only patients concomitantly treated with selective serotonin reuptake inhibitors (SSRI) improved. In another open-label study, nine patients with AN received 20 sessions of tDCS (the anode over the left DLPFC and the cathode over the right DLPFC). The treatment was effective on several AN dimensions and the comorbid depression at 1 month of follow-up [56]. Finally, a pilot study is currently testing the efficacy and safety of high-definition tDCS over the left inferior parietal lobe [57].

In BN and BED, suppression of the urge to binge eat and increased self-regulatory control were reported by the only double-blind sham-controlled proof-of-principle trial on the effects of bilateral tDCS over the DLPFC in 39 adults with BN [59]. Two electrode montages were tested: anode on the right and cathode on the left (AR/CL) and the reverse (AL/CR). One session of AR/CL led to bulimic cognition reduction and mood improvement, compared with the AL/CR and sham conditions [59]. A case report [60] and the study by Burgess et al. (*n* = 30 participants with BED) showed that compared with the sham arm, tDCS on the right DLPFC decreased cravings for sweets, savory proteins, and an all-food category with stronger reductions in men. A possible explanation of this finding is that stimulation of the right DLPFC enhances cognitive control and/or decreases the need for reward [58].

### 3.3. ECT in Eating Disorders 

#### 3.3.1. Anorexia Nervosa

As shown in Table 3, only case reports and case series have evaluated ECT in AN (*n* = 50 patients in total) [61]. In most cases, ECT was proposed to treat a comorbid major depressive disorder (MDD), especially treatment-refractory MDD or associated with suicide attempts or obsessive-compulsive disorder (OCD). Indeed, due to the high suicide rate in patients with EDs, intensive treatments, like ECT, are often proposed. The recent and largest case series on patients with AN treated by ECT concerned 30 adolescents with comorbid depression [62]. All were severely depressed and suicidal upon admission and resistant to antidepressants. Improvement in depressive and ED symptoms was observed after ECT with minimal adverse effects. Several years after discharge, 46.6% of patients had no evidence of depression, suicidality, and ED symptomatology, and 23% had only ED symptomatology. Due to the simultaneous improvement of AN symptoms, body weight, and depression, a specific effect on ED cannot be easily identified. Finally, two recent case reports show contradictory results of ECT in patients with AN without psychiatric comorbidity [63,64]. Specifically, Naguy et al. demonstrated a strong improvement of eating behaviors in a 16-year-old girl after only six ECT sessions. However, the absence of comorbid depression or OCD was not clearly stated, and high-dose antidepressant treatment was introduced during the ECT course. Duriez et al. reported that 10 sessions of ECT had a negative outcome and rapid relapse after discharge in patients with careful evaluation of depression, anxiety, and ED dimensions before and after ECT.

To our knowledge, no controlled trial is currently testing ECT efficacy in AN. Furthermore, as only case series have been published, no standardized evaluation has been provided, thus preventing the precise comparison of patients with different AN profiles. Moreover, we hypothesize a strong publication bias towards ECT in severe and enduring AN without comorbid psychiatric disorders.

Nevertheless, the available results suggest that ECT is safe in this population and might improve AN symptomatology in addition to its positive effects on comorbid mood disorders. ECT appears to be well tolerated in this population, and it is even used in late-onset AN in older adults [67,70].

#### 3.3.2. Bulimia Nervosa and Binge Eating Disorder

MDD is more frequent in patients with BN than in AN [77]. According to a recent systematic review [61], only one case report described ECT use in BED and none in BN. In this patient with a psychiatric comorbidity, ECT was safe and effective on bulimic symptoms.

### 3.4. VNS in Eating Disorders

To date, no study assessed VNS effects in patients with EDs [78]. However, a growing body of evidence suggests the relevance of VNS in patients with ED. Some studies in animal models showed an association between VNS and reduction in food intake and/or weight loss, suggesting that vagal stimulation might mediate satiety signals (for review see McClelland 2013 [79]). Several fMRI studies have also shown that VNS modulates the activity in brain regions related to the processing of afferent vagal signals and interoception, such as the thalamus, precentral gyrus, and insular cortex [80,81,82]. A recent study demonstrated that transcutaneous VNS improves interoceptive accuracy [83]. This is a very valuable point given the central role of interoception in ED [84,85,86].

### 3.5. Deep Brain Stimulation in Eating Disorders

#### 3.5.1. Deep Brain Stimulation in Anorexia Nervosa

In a recent review, Sobstyl et al. described all trials on DBS in patients with AN [87]. Ten years ago, the first description of DBS in a patient with severe and treatment-resistant MDD and comorbid AN opened the way to a new therapeutic modality [88]. To date, 58 case reports on DBS in AN have been published (Table 4). The first studies concerned well known targets in MDD or OCD, mostly in the limbic system linked to the anxiety and emotion pathways. Barbier et al. [89] described AN remission and complete weight recovery in a patients with comorbid OCD/AN after treatment first by bilateral stimulation of the anterior limb of the internal capsule and then of the bed nucleus of the stria terminalis. In 2012, McLaughlin et al. reported the improvement of severe OCD in a patient with comorbid AN treated by DBS of the ventral capsule and ventral striatum [90]. Moreover, they observed a small weight gain, but less distress about caloric intake. Blomstedt et al. [91] showed that bilateral stimulation of the medial forebrain bundle (MFB) and then of the bed nucleus of the stria terminalis (BNST) improved MDD and AN. However, it should be noted that this 60-year-old woman lost weight during the procedure. The first case series came from an open-label trial involving 16 patients with AN, among whom 14 had comorbid depression or other major psychiatric comorbidities [92]. The primary outcomes were acceptability and safety of DBS applied to the subgenual cingulate cortex (SCC), a validated target for the treatment of resistant depression [93]. After 1 year, patients exhibited weight gain and improvement in depressive and anxious symptoms. In 2013, AN remission was observed in four patients after implanting electrodes bilaterally in the nucleus accumbens (NAcc) [94]. Wang et al. [95] used the same method in two patients with AN and a comorbid psychiatric disorder (OCD, generalized anxiety disorder, or MDD). These two trials also demonstrated glucose metabolism changes after DBS in the NAcc and SCC by positron emission tomography [96,97]. Recently, the Shanghai group published the largest series on DBS in AN involving 28 patients with refractory AN who were followed for at least 2 years after electrode implantation in the NAcc [98]. All patients had a major psychiatric comorbidity at inclusion (*n* = 9 OCD, *n* = 7 severe anxiety, and *n* = 12 MDD). Post-hoc analysis suggests that NAcc DBS is less effective for weight restoration in the binge/purge AN subtype than in the restrictive subtype [98]. Another recent preliminary study proposes for the first time two targets in the same trial chosen according to the main psychiatric comorbidities associated with AN: SCC for affective disorder (*n* = 4) and NAcc for anxiety disorder (*n* = 4) (Martinez et al., 2020). Four patients considered as responders after 6 months will be randomized in two arms (ON/OFF or OFF/ON) for a double-blind controlled cross over trial.

Regarding tolerance and safety, DBS is a reversible procedure, and the device can be removed if requested by the patient [93,96]. The clinical situation of malnutrition is a key issue for surgical procedures, like DBS [92]. The existing studies did not report any permanent neurological deficit after the procedure. In one patient, the device was removed due to infection (new implantation 6 months later). Moreover, Lipsman et al. observed excessive pain at the incision site in five patients, treatment withdrawal by two patients (device off or removal) without any precise reason, and seizures in two patients. Liu et al. reported device removal in one patient (3%) at 18 months due to rejection. It has been suggested that the surgical risk is higher in patients with severe AN and BMI lower than 14 kg/m^2^ due to malnutrition. These first findings are encouraging, but more investigations and controlled trials are needed. Nevertheless, the high mortality and morbidity in severe and enduring AN and the increasing knowledge of AN functional neuroanatomy give strong ethical support for this procedure.

#### 3.5.2. Deep Brain Stimulation in Bulimia Nervosa and Binge Eating Disorder

As shown in Table 4, several patients have been treated by DBS of the hypothalamus or NAcc for severe obesity [105]. In some cases, binge eating behaviors were mentioned, but without a clear ED diagnosis. A pilot study with a 2-year follow-up assessed DBS of the lateral hypothalamic area in three patients with refractory obesity [103]. Only one patient reported binge eating reduction. Tronier et al. observed a reduction of binge eating behaviors after bilateral DBS of NAcc in one patient with treatment-resistant depression and severe obesity previously treated by gastric bypass [104].

#### 3.5.3. Neuroanatomical Targets in DBS

SCC was the first DBS target tested in AN by Andres Lozano’s group after their extensive experience with DBS in depression. This region is an affective regulatory center [106]. Substantial evidence indicates the main role of dysregulated emotional processing in AN pathogenesis [107]. SCC stimulation in AN also benefits from the experience in refractory MDD, and is an extensively interconnected component of the limbic system.

Other limbic regions have also been targeted. The BNST is a center of integration for limbic information and valence monitoring [108]. The MFB is a key structure of the reward-seeking circuitry and is highly connected to the limbic system [109]. Preclinical data supports the interest of NAcc modulation in EDs and the stimulation of the ventral striatum. NAcc stimulation increased food intake and weight gain in a rodent model of food restriction and hyperactivity [110,111]. Abnormalities in goal-directed behavior and the establishment of a compulsive/restrictive behavior might be the consequences of dysregulation of neurocircuits that control positive/negative valence as well as reward and decision-making behaviors [112]. A recent study highlighted the role of the mesolimbic reward circuitry in a rodent model of AN [113].

It is important to stress that when choosing targets for brain stimulation, the dysfunctional networks in EDs must be taken into account, as illustrated by recent studies on DBS. Lipsman et al. and Zhang et al. observed broad changes in glucose metabolism in many brain regions after DBS of the SCC and NAcc, respectively. Specifically, SCC stimulation increases activity in the insula and glucose metabolism in parietal and temporal regions, and decreases cingulate activity [93]. NAcc stimulation decreases activity in the frontal lobe, lentiform nucleus (putamen), and hippocampus [97]. More studies are needed to investigate the influence of ventral striatum (NAcc) stimulation and dorsal striatum (caudate nucleus and putamen) on habit formation. The DBS mechanism of action on neuronal pathways is only partially understood, but the target choice is the focus of the current research because a highly focal intervention can have a very broad effect.

## 4. Discussion

### 4.1. Current Evidence and Issues

Altogether, the evidence for the use of brain stimulation in EDs is promising, but more studies are needed before it will be considered an effective intervention. Indeed, the literature consists mainly of case reports, cases series, and open studies, and only a few randomized controlled trials. Most studies had small samples and focused mainly on the immediate effects on craving or neurocognition, without follow-up data. Moreover, methodologies were very heterogeneous among studies. For instance, some studies on rTMS effects on craving did not use cues to induce craving, while others used them only during stimulation or for pre-stimulation craving induction, but not for post-stimulation craving assessment. Some studies assessed craving using a visual analogic scale, whereas other used questionnaires. Moreover, only a few studies assessed the (immediate and long-term) clinical effects of brain stimulation. Yet, a crucial question is whether brain stimulation can induce lasting changes in a well-established behavior. Studies using ECT are limited to case reports. Nevertheless, there have been more studies on ECT in AN in the past two years than in the previous thirty years. This is probably due to a global increase of interest in ECT after years of stigma and the recent demonstration of neurogenesis induction by ECT [114]. Although the level of evidence is low, ECT may be useful and safe for the management of severe and treatment-resistant MDD in patients with AN. Its wider accessibility compared to other brain stimulation techniques should facilitate the organization of a prospective trial, particularly in patients with severe comorbid MDD, given the lower efficacy of SSRI in AN [5]. The cognitive and memory effects of ECT are a challenging aspect. They are the strongest limiting factors of ECT use in MDD and should be thoroughly evaluated in the specific metabolic context of AN. DBS is the most recent and promising brain stimulation technique for severe and enduring AN (due to the ethical issues linked to an invasive procedure). Although the lack of consensus on the best neuroanatomical target strongly limits the level of evidence, there are already prospective studies and a few randomized trials currently recruiting. Park et al. proposed a double-blind cross-over study that includes a sham-stimulation phase [92]. Perez et al. are currently recruiting for a cross-over trial in Spain (NCT03168893) [102].

The main issue of the reviewed studies is that many of them were underpowered. Therefore, their findings must interpreted with caution due to the high risk of type II error and inflated effect sizes. Future studies should include larger samples, and the number of patients needed for a robust statistical analysis should be calculated in advance. In rTMS, most studies used manual methods rather than MRI-based methods to locate the target, particularly those on rTMS of the DLPFC. As the location of the intended target region varied across individuals, this might have affected the results and resulted in low effect-size. Moreover, the standard figure-8-shaped rTMS coil, used in most studies, allows a relatively limited and shallow stimulation area that does not induce direct stimulation of deep cortical areas [115]. Stimulation of deeper areas using an H coil might be more effective, leading to enduring benefits. With the exception of Dalton et al. [28,44], the randomized controlled studies only proposed a limited number of pulses and 1 to 10 sessions (i.e., about 10,000 pulses maximum). It is likely that more sessions are needed to modify neural programs and their associated behaviors. Future studies should determine the optimal dose of neuromodulation and duration (for example by comparing different rTMS durations or by investigating in ancillary studies non-responders to a specific program).

Psychiatric comorbidities are the norm in people with eating disorders (>70%) [1]. Binge disorders are often comorbid with a substance use disorders [1]. Binge eating is also frequently compared to addictions, based on the evidence that they share common characteristics, such as escalating frequency of the behavior, ambivalence towards treatments, and frequent relapses [116,117]. In addition, brain stimulation is a successful strategy in MDD. ECT is recommended in most guideline and rTMS in some of them [118]. Given the high rate of these comorbid disorders in patients with EDs, some of the published studies were built on this background. Protocols targeting DMPFC in BN were adapted from substance use disorders studies, ECT studies on severe MDD have been one of the main drivers to begin ECT in patients with MDD and AN. In DBS, AN comorbid with MDD and OCD was one of the reasons to target the SCC. In their study targeting two areas in the same trial for the first time, the targets were chosen based on two of the major psychiatric comorbidities associated with AN: SCC for affective disorder (*n* = 4) and NAcc for anxiety disorder (*n* = 4) [102]. Indeed, most studies reported positive effects on depressive symptoms with rTMS [26], tDCS [56], ECT [62], and DBS [93]. Future studies in EDs might benefit from this knowledge in brain target selection (Figure 1), in studies design (add-on with another modalities of treatment, inclusion and exclusion criteria, etc.), or in the stimulation parameters: number of sessions, type of coil or electrode, stimulation duration, etc. Nevertheless, regarding the mood component, there is a crucial methodological problem: because of the simultaneous improvement of ED weight and mood features, a specific effect on ED cannot be identified. Systematic measurement of depressive symptoms associated with subgroup analyses of patients without depression will make it possible to address this problem.

Another factor that should be considered is the variability in brain activity related to the metabolic state. The nutritional state in patients with ED might affect brain functions more than in any other psychiatric disorder. Starvation and nutritional status affect behavior, cognition, and disease symptoms. The nutritional status also influences the treatment response, especially to antidepressants that are less effective in acutely ill and underweight patients [5]. To the best of our knowledge, no study has determined whether the nutritional status affects the resting brain state and the neuromodulatory response. Nevertheless, it is possible that the nutritional status influences the response/non-response to a treatment and acts as a confounding factor on brain stimulation efficiency. Better metabolic monitoring could be useful to limit some central deficit that might affect the stimulation response, as reported for folate deficiency in people with comorbid depression [119] and for kynurenic pathway defects [120]. This should be investigated in view of personalized treatment programs.

Finally, in most of the studies reviewed here, samples only included adults and mainly patients with severe, chronic ED. However, due to the good safety and acceptability profile of NIBS, it would be interesting to assess their effects in patients with less severe ED forms, as is usually done in studies on SSRI efficiency in bulimia [121].

### 4.2. Perspectives

#### 4.2.1. Improving Brain Stimulation Efficiency

A promising strategy consists in stimulating a specific disorder-related circuitry involved in ED using NIBS and in functionally engaging the targeted circuit through cognitive tasks or therapies. This is particularly relevant for neurocognition.

Behavioral abnormalities driven by cognitive processes are a prominent ED feature. For instance, studies in adults with AN suggest poor set-shifting [122], weak central coherence [123], and impaired decision making [124]. Similarly, in BN and BED, decision making and cognitive inhibition seem to be impaired [94,124]. Cognitive remediation programs that target specifically some of these functions have been developed, and some have proved their effectiveness, notably in AN [125]. NIBS also may modulate some of these cognitive functions [126]. It would be interesting to determine whether concomitant brain stimulation might enhance cognitive remediation training. This is especially true for tDCS, which is easier to administer and does not interfere with the psychotherapy sessions. Some proof of concept studies are currently testing them in EDs [127]. Moreover, studies on other disorders showed that when a patient experiences salient cues before modulation with TMS, the benefits increase significantly compared with patients who receive no provocation task [128]. An optimized provocation task before or during brain stimulation is one of the future challenges. In this context, the widespread availability of virtual reality gaming environments that could be customized and adapted to EDs (for example with cues related to food or body shape) might be an opportunity.

In DBS, the stimulation parameters also may be critical. This includes frequency, voltage, pulse width, and also number of contacts and number of directions depending on the electrode model. The choice of parameters remains challenging [129] because symptom improvement cannot be immediately monitored, in contrast to movement disorders. None of the reviewed studies on DBS evaluated the stimulation parameters, although this is a crucial issue. Awake surgery, proposed by Lipsman to increase the procedure safety, could be used also to optimize the target location by testing during implantation [93]. Alternatively, deep electroencephalography during a neurocognitive task immediately before electrode positioning may help to precisely choose the target. In addition, in the currently used open-loop DBS, the medical team can modify the stimulation parameters at different times based on clinical changes. In closed-loop DBS, the stimulation parameters are programmed automatically based on the measured biomarker. Closed-loop devices could adapt the stimulation as a function of the eating behaviors during food consumption or food deprivation. External interventions can modulate some therapeutic strategies to model behaviors like exposition work during specific cognitive behavioral therapy. Finally, the current brain stimulation strategies have little access to important therapeutic targets, deep within the brain. Recent NIBS studies [130,131] in rodent models suggest that deep-brain areas could be targeted in a non-invasive manner, potentially enlarging the number of patients who might benefit from deep-brain stimulation due to the reduction of risk and the perspective of mapping deep-brain targets. Currently, this promising technique has only shown its effectiveness on rodent brains. Its relevance and spatial resolution in human brains, which are much larger, are unknown. Resolution might be increased also using optogenetic techniques, but these methods raise ethical questions linked to their translation from animals to humans [132,133].

#### 4.2.2. Toward More Personalized Protocols

As EDs and each subtype are highly heterogeneous and have many overlapping features, finding a single “optimal” protocol is highly unlikely. Moreover, it has been suggested that there are distinct neural endophenotypes, not readily apparent when using standard diagnostic criteria, but with differential neural and clinical responses to interventions.

Targeting specific subgroups is a valuable option. For instance, many findings support a specific disease trajectory, and preliminary evidence suggests that interventions should be matched to the disease stage [134]. Interventions tailored according to ED stage or to the developmental trajectory were tested in a recent pilot study [30]. Some clinical features could also be considered because they are associated with a specific disease form or poorer prognosis, such as ED subtype (restrictive versus binge), age, gender, associated personality traits, psychiatric comorbidities, or history of childhood abuse. Latent class and profile analyses have been performed and could be a starting point for patient stratification.

Another possible solution is to move from a categorical to a dimensional approach and to target relevant dimensions associated with the disorders rather than with the specific diagnosis. This might be particularly relevant within the research domain criteria (RDoC) research framework. This project seeks to characterize the fundamental domains of cognitive, perceptual, and social processing with the aim of identifying novel targets for mental health disorder treatment. It integrates many levels of information (from genomics and circuits to behavior and self-reports) to assess basic dimensions of functioning that span the full range of human behavior from normal to abnormal. These basic dimensions, independent of DSM diagnoses, are then used to describe the pathological behaviors of psychiatric disorders. Many dimensions associated with EDs can be studied within the RDoC matrix, such as delay discounting, sensitivity to reward/punishment, compulsive behavior, and cognitive functions [135,136]. Brain stimulation is a tremendously interesting approach to modulate the cerebral circuits involved in these dimensions and to assess the impact of this modulation in an integrative way from the molecular level to behavior.

The effects of a given protocol can vary widely across individuals. Some patients will drastically improve, whereas others will not improve, or will even worsen. ED heterogeneity and the potential confounding factors might play a major role, and therefore it is crucial to identify predictors and correlates of the response to a treatment. The perfect illustration of this is the study by Dunlop et al. [40] where enhanced frontostriatal connectivity was associated with response to DMPFC rTMS for binge/purge behavior. rTMS caused paradoxical suppression of frontostriatal connectivity in non-responders who had a high connectivity on resting-state fMRI. Thus, resting-state fMRI could be a key tool to optimize the stimulation parameters. Moreover, in a study on cocaine users treated by continuous theta-burst stimulation of the medial prefrontal cortex, the effects on the neural circuitry of cravings were not uniform and may depend on the individual baseline frontal-striatal reactivity to cues [137]. Hayes et al. also demonstrate that presurgical fornix and anterior limb of internal capsule (ALIC) connectivity are correlated with DBS response [100]. If these results are replicated, this feature might be useful in selecting potential responders. These examples indicate that future studies should include as secondary objectives the profiles of responders/non-responders. It also illustrates the relevance of fMRI neuromodulation studies for patient selection and for investigating the neuromodulatory mechanism. Martinez et al. adapted the DBS target depending on the main comorbidity in patients with severe and resistant AN: SCC for affective disorder or NAcc for anxiety disorder [102]. Finally, all DBS studies defined a minimal illness duration (from 7 to 10 years) as eligibility criterion. Better evidence for choosing the criteria of ED severity, based on neurocognitive tasks, neuroimaging, neuronal metabolism, genetic, or other metabolic patterns, is needed to allow evaluating DBS in patients with shorter illness duration [138]. More precise neurocognitive analyses correlated with neurophysiological measures are also needed.

## 5. Conclusions

Brain stimulation techniques in EDs are still in their infancy. Different potential targets have been considered (Figure 1), but the literature includes mainly open studies with only a few randomized controlled trials, and many issues remain to be addressed. At this time, the evidence for using brain stimulation as a routine treatment in ED management is weak. However, several ongoing studies should bring new information on optimal stimulation protocols and targets. This is a major source of hope for EDs where the development of alternative treatments is crucial.

## Figures and Tables

**Figure 1 jcm-09-02358-f001:**
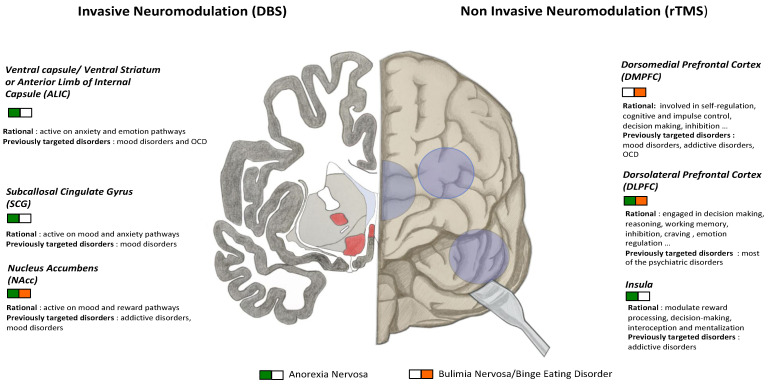
Overview of main neuromodulation targets in eating disorders. Schematic coronal section of right brain and front view of left brain. DBS: deep brain stimulation; rTMS: repetitive transcranial magnetic stimulation; tDCS: transcranial direct current stimulation; OCD: obsessive-compulsive disorder.

**Table 1 jcm-09-02358-t001:** Repetitive transcranial magnetic stimulation (rTMS) in eating disorders.

Reference	Type of Study	Participants	Modulation Target	Treatment Characteristics	Main Results
Anorexia Nervosa					
Kamolz et al., 2008 [19]	Case report	24-year-old female with AN	Left DLPFC Manual targeting	41 sessions 100 × 2 s trains/10 s inter-train interval at 10 Hz = 2000 pulses per session, 110 % MT	Full remission
Van den Eynde et al., 2013 [24]	Case series (pilot study)	10 patients with AN	Left DLPFC Manual targeting	1 session, 20 × 5 s trains/55 s inter-train interval, 10 Hz, intensity of 110% MT, 1000 pulses over 20 min	Reduced levels of feeling full, feeling fat, and feeling anxious
McClelland et al., 2013 [20]	Case report	23-year-old and 52-year-old women with AN ^1^	Left DLPFC Neuronavigation	20 and 19 sessions 20 × 5 s trains/55 s inter-train interval, 10 Hz, intensity of 110% MT, 1000 pulses over 20 min within each session	Significant improvement
McClelland et al., 2016 [23]	Case series	5 women with AN ^1^	Left DLPFC Neuronavigation	~20 sessions, 20 × 5 s trains/55 s inter-train interval, 10 Hz, intensity of 110% MT, 1000 pulses over 20 min within each session	Significant improvement of ED and affective symptoms after 6 months, but positive results waned at 12 months follow-up
McClelland et al., 2016 [25]	RCT	49 patients with AN ^1^	Left DLPFC Neuronavigation	1 session 5 s trains/55 s inter-train interval, 10 Hz, intensity of 110% MT, 1000 pulses over 20 min	No significant effect on core symptoms of ED compared to sham rTMS, but improvement in individuals who received real rTMS when compared before and after the session and the results persisted at 24 h of follow-up
Choudhary et al., 2017 [21]	Case report	23-year-old female with AN	Left DLPFC Manual targeting	21 sessions High-frequency rTMS (10 Hz) at 110% of resting MT, 1000 pulses	Significant improvement
Jaššová et al., 2018 [22]	Case report	25-year-old female with AN	Left DLPFC Targeting method not available	10 sessions, 10 Hz, 15 trains/day, 100 pulses/train, intertrain interval 107 s	No improvement of ED, anxiety, or depression
Woodside et al., 2017 [29]	Case series	Fourteen subjects with eating disorders (6 AN, 5 BN, and 3 ednos) and comorbid PTSD	DMPFC Neuronavigation	20–30 sessions 120% resting MT, at 10 Hz, 5 s on, 10 s off, 3000 pulses/hemisphere, with left then right lateralized coil orientation	Improvement in emotional regulation and PTSD symptoms
Dalton et al., 2018 [26]	RCT	30 patients (16 real, 14 sham) with AN ^2^	Left DLPF CNeuronavigation	20 sessions, 20 × 5 s trains/55 s inter-train interval, 10 Hz, intensity of 110% MT, 1000 pulses over 20 min within each session	Between-group effect sizes of change scores (baseline to follow-up) were small for BMI (d = 0.2, 95% CI −0.49 to 0.90) and eating disorder symptoms (d = 0.1, 95% CI −0.60 to 0.79), medium for quality of life and moderate to large (d = 0.61 to 1.0) for mood outcomes, all favoring rTMS over sham
Knyahnytska et al., 2019 [30]	Case series (pilot study)	8 women with AN	Insula Manual targeting	42 sessions, H-coil dTMS 18 Hz, 2 s on, 20 s off, number of pulses 36, number of trains 80, over 20 min	Reduction in AN-related obsessions and compulsions, as well as depression and anxiety scores
Dalton et al., 2020 [28]	RCT	34 anorexic female patients (17 real, 17 sham) vs. 30 healthy controls ^2^	Left DLPFC Neuronavigation	20 sessions 20 × 5 s trains/55 s inter-train interval, 10 Hz, intensity of 110% MT, 1000 pulses over 20 min within each session	No significant effect of rTMS nor time on food choices related to fat content. Among AN participants who received real rTMS, there was a decrease in self-controlled food choices at post-treatment
Bulimic Disorders (Bulimia and/or Binge Eating Disorders)					
Hausmann et al., 2004 [31]	Case report	One woman with BN and depression	Left DLPFC Neuronavigation	10 sessions, 20 × 5 s trains/55 s inter-train interval, 10 Hz, intensity of 110% MT	Significant improvement in BN symptoms
Walpoth et al., 2008 [32]	RCT	14 females with BN	Left DLPFC Manual targeting	15 sessions, 10 × 10 s trains/60 s inter-train interval at 20 Hz, = 2000 pulses; 120% MT	No difference between real and sham group
Van den Eynde et al., 2010 [33]	RCT	38 females with BN ^3^	Left DLPFC Manual targeting	1 session, 20 × 5 s trains/55 s inter-train interval, 10 Hz, intensity of 110% MT, 1000 pulses over 20 min	Real rTMS associated with a decrease in self-reported urge to eat and binge eating
Van den Eynde et al., 2011 [34]	RCT	33 females with BN ^3^	Left DLPFC Manual targeting	1 session, 20 × 5 s trains/55 s inter-train interval, 10 Hz, intensity of 110% MT, 1000 pulses over 20 min	No differences between the real and sham groups on stroop task
Van den Eynde et al., 2011 [35]	RCT	38 females with BN ^3^	Left DLPFC Manual targeting	1 session, 20 × 5 s trains/55 s inter-train interval, 10 Hz, intensity of 110% MT, 1000 pulses over 20 min	TMS did not alter blood pressure or heart rate
Claudino et al., 2011 [36]	RCT	22 patients (11 real, 11 sham) with BN ^3^	Left DLPFC Manual targeting	1 session, 20 × 5 s trains/55 s inter-train interval, 10 Hz, intensity of 110% MT, 1000 pulses over 20 min	Decreased salivary cortisol concentrations compared with sham rTMS
Van den Eynde et al., 2012 [37]	Case series	7 left-handed females with BN	Left DLPFCM anual targeting	1 session, 20 × 5 s trains/55 s inter-train interval, 10 Hz, intensity of 110% MT, 1000 pulses over 20 min	Different effects in left- and right-handed people
Downar et al., 2012 [38]	Case report	One woman with severe refractory BN and depression	DMPFC Neuronavigation	2 × 20 sessions 60 trains of 10 Hz stimulation at 120% of resting motor threshold in 5 s trains with a 10-s inter-train interval	Significant improvement in BN symptoms
Baczynski et al., 2014 [39]	Case report	One woman with BED and comorbid depression	Left DLPFC Manual targeting	20 sessions 20 × 4 s trains/26 s inter-train interval, 10 Hz, intensity of 110% MT	Improvement in binge eating scale
Dunlop, 2015 [40]	Case series	28 subjects with anorexia nervosa, binge-purge subtype or bulimia nervosa	DMPFC Neuronavigation	20–30 sessions 120% resting MT, at 10 Hz, 5 s on, 10 s off, 3000 pulses/hemisphere, with left then right lateralized coil orientation	Enhanced frontostriatal connectivity was associated with responders to DMPFC-rTMS for binge/purge behavior
Sutoh et al., 2016 [41]	Case series (pilot study)	8 women with BN	Left DLPFC Manual targeting	1 session, 15 × 5 s trains/55 s inter-train interval, 10 Hz, intensity of 110% MT, 1000 pulses over 20 min within each session	Significant reduction of food craving and decrease in cerebral oxygenation of the left DLPFC
Gay et al., 2016 [42]	RCT	47 women (23 real, 24 sham) with BN ^4^	Left DLPFC Manual targeting	10 sessions, 20 × 5 s trains/55 s inter-train interval, 10 Hz intensity of 110% MT, 1000 pulses over 20 min within each session	No significant improvement of bingeing or purging
Guillaume et al., 2018 [43]	RCT	39 patients (22 real, 17 sham) with BN ^4^	Left DLPFC Manual targeting	10 sessions, 20 × 5 s trains/55 s inter-train interval, 10 Hz, intensity of 110% MT	Improvement of inhibitory control and decision-making

Note: All studies are cited in bibliography, because when a recent systematic review was available, it was included and prioritized in the synthesis of results. ^1,2,3,4^: Partial or total overlap on sample.

**Table 2 jcm-09-02358-t002:** Transcranial direct current stimulation (tDCS) and eating disorders.

Reference	Type of Study	Participants	Modulation Target	Treatment Characteristics	Main Results
Khedr et al., 2014 [55]	Open-label, single arm study (pilot study)	7 patients with AN	Left DLPFC	10 sessions Anodal tDCS for 25 min at 2 mA (15 s ramp in and 15 s ramp out)	Immediate improvement in 3 patients after the sessions on eating and depressive symptoms, with one patient maintaining the response at 1 month
Burgess et al., 2016 [58]	RCT (proof-of-concept study)	30 participants with BED	Right DLPFC	1 session, 2 mA, 20 min	Decreased craving for sweets, savory proteins, and an all-foods category, with strongest reductions in men
Kekic et al., 2017 [59]	RCT	39 patients (2 male) with BN	Right and left DLPFC (3 montages: AR/CL; AL/CR; sham)	1 session 20 min, 2 mA, 10 s ramp on/off	Reduction in ED cognitions with AR/CL tDCS Suppression of urge to binge-eat and increased self-regulatory control with both active montages Improvement of mood with AR/CL
Sreeraj et al., 2018 [60]	Case report	37 year old female with schizophrenia and binge-eating	Right DLPFC	10 sessions, 2 mA, 30 min	Improvement in subjective reporting on cognitive restraint and control over eating as well as feeling of satiation and ability to eat after exposure to cues 3 kg weight loss by the end of the treatment, 7 kg at 10-month follow-up
Strumila et al., 2019 [56]	Open-label study	10 females with AN	Anode over left DLPFC and cathode over right DLPFC	20 sessions of anodal 2 mA stimulation during a period of two weeks	Improvement of anorexic and depressive symptoms

**Table 3 jcm-09-02358-t003:** Electroconvulsive therapy (ECT) and eating disorders.

Reference	Type of Study	Participants	Modulation Target	Treatment Characteristics	Main Results
Anorexia Nervosa					
Davis et al., 1961 [65]	Case report	12-year-old girl with AN-R	-	12 sessions (bilateral)	Weight gain and discharge
Bernstein et al., 1964 [66]	Case report	20-year-old female with AN-R and personality disorder	-	21 sessions followed by maintenance ECT	Weight gain and mood improvement
Bernstein et al., 1972 [67]	Case report	94-year-old female with AN-R and psychotic disorder	-	5 sessions	Short term weight gain
Ferguson et al., 1993 [68]	Case series	3 patients with AN and MDD	-	11, 8 and 16 sessions (bilateral)	Transient improvement on weight and symptomatology for 2/3 patients
Bek et al., 1996 [69]	Case series	8 females with AN, one had psychosis and five had personality disorders	-	11 sessions	Modest weight gain
Hill et al., 2001 [70]	Case report	77-year-old female with AN-R and MDD	-	9 sessions	Modest weight gain and mood improvement
Poutanen et al., 2009 [71]	Case report	21-year-old female with AN-B/P and MDD	-	45 sessions in three courses (bilateral)	Modest thymic and eating amelioration. Cognitive impairment.
Andrews et al., 2014 [72]	Case report	17-year-old with AN-B/P, MDD, and NSSI	-	10 unilateral and 21 bilateral sessions/13 weeks	Mood improvement
Andersen et al., 2017 [73]	Case report	14-year-old girl with AN-R, MDD, and GAD	-	22 sessions (bilateral)	Weight gain
Saglam et al., 2018 [74]	Case report	24-year-old male with AN-B/P, OCD, and MDD	-	12 sessions (bitemporal)	Weight restoration and OCD improvement, stopped diuretic and laxative abuse.
Pacillio et al., 2019 [61]	Case report	30-year-old female patient with AN and MDD	-	11 sessions (unilateral)	Modest increase of eating disorder, mood improvement
Naguy et al., 2019 [63]	Case report	16-year-old female with AN and personality disorder	-	6 sessions (bitemporal)	Weight gain and improvement in eating behavior
Duriez et al., 2020 [75]	Case report	19-year-old female with AN	-	10 sessions	No improvement of AN symptoms
Shilton et al., 2020 [62]	Case series	30 female adolescents with AN and MDD	-	-	Mood improvement, treatment well tolerated, no specific improvement for eating disorder symptoms
Bulimic Disorders (Bulimia and/or Binge Eating Disorders)					
Rapinesi et al., 2013 [76]	Case report	41-year-old male with BED and bipolar disorder. Personal history of AN	-	8 sessions (bitemporal)	Important weight loss and decrease of psychotic symptoms

AN: anorexia nervosa; AN-R: anorexia nervosa restricting subtype; AN-B/P: anorexia nervosa binge/purge subtype; BED: binge eating disorder; OCD: obsessive-compulsive disorder; MDD: major depressive disorder; NSSI: non-suicidal self-injury.

**Table 4 jcm-09-02358-t004:** Deep brain stimulation (DBS) in eating disorders.

Reference	Type of Study	Participants	Modulation Target	Treatment Characteristics *	Main Results
Anorexia Nervosa					
Israël et al., 2010 [88]	Case report	56-year-old female with AN and severe depression	SCC (bilateral)	Intermittent stimulation 2 min on/1 min off 5 mA/91 µs/130 Hz	Maintenance of normal BMI (average 19.1 kg/m^2^) at 3 years, normal scores in restraint and weight and shape concerns
Barbier et al., 2011 [89]	Case report	39-year-old female with AN and severe OCD	ALIC and BNST (bilateral)	Unknown	Full recovery of AN and strong improvement of OCD
McLaughlin et al., 2012 [90]	Case report	52-year-old female with refractory OCD and AN	Ventral capsule and ventral striatum (bilateral)	Left unilateral, monopolar 7.5 V/120 µs/120 Hz	Significant weight improvement, reduction in AN-related obsession and patient can go out to eat
Wu et al., 2013 and Sun et al., 2012 [94,99]	Case series	4 females with AN (3 OCD, 1 GAD) ^1^	NAcc (bilateral)	Unknown	Full remission of AN, restoration of menstrual cycle and return to school for 3 patients
Wang et al., 2013 [95]	Case series	2 females with AN, depression, and OCD	NAcc (bilateral)	2.5–3.8 V/120–210 µs/135–185 Hz	Significant weight gain and affective improvement
Lipsman et al., 2013 [96]	Open label clinical trial	6 females with AN, 5 with psychiatric comorbidities (MDD, OCD, SUD, PTSD) ^2^	SCC (bilateral)	5–7 V/90 µs/130 Hz	Weight gain in 3 patients, changes in brain metabolism
Hayes et al., 2015 [100]	Ancillary Study	8 females with AN, 7 with psychiatric comorbidities (MDD, OCD, GAD, PTSD, BPD) ^2^	SCC (bilateral)	Unknown	Weight loss in 3 patients, weight gain in 5 patients
Lipsman et al., 2017 [93]	Open label clinical trial	16 females with AN, 14 with psychiatric comorbidities (MDD, OCD, SUD, PTSD, GAD, BPD) ^2^	SCC (bilateral)	5–6.5 V/90 µs/130 Hz	Significant weight gain for 8 patients Adverse effects: 1 surgical-site infection, 2 devices explanted at patient request, 1 seizure
Blomstedt et al., 2017 [91]	Case report	60-year-old female with AN and depression	MFB (bilateral) and subsequent BNST (bilateral)	Bipolar MFB stimulation 3 V/60 µs/130 Hz two years later: monopolar BNST stimulation 4.3 V/120 µs/130 Hz	Improvement of affective symptoms Weight stabilization Target change due to blurred vision
Manuelli et al., 2019 [101]	Case report	37-year-old female with AN-BP	BNST (bilateral)	4 V/60 µs/130 Hz	Full weight restoration after 4 months
Wei Liu et al., 2020 [98]	Open label clinical trial	29 females with AN, 28 with psychiatric comorbidities (12 MDD, 9 OCD, 7 GAD)	NAcc (bilateral)	2.5–4 V/120–150 µs/160–180 Hz	12 patients obtained full weight restoration and 5 significant weight increase after 2 years of follow up Less effective with AN-BP than AN-R
Martinez et al., 2020 [102]	Open label clinical trial ^3^	7 female and 1 male with AN, 4 with affective disorder and 4 with anxiety disorder as main psychiatric comorbidities	SCC (bilateral) or NAcc (bilateral)	7–8 mA/90 µs/130 Hz	No weight gain. Subjective improvement of quality of life
Bulimic Disorders (Bulimia and/or Binge Eating Disorders)					
Whiting et al., 2013 [103]	Case series	3 patients with BED	LHA (bilateral)	Monopolar unknown V/90 µs/185 Hz	1/3 significantly improvement in binge eating Significant weight loss in 2/3
Tronnier et al., 2018 [104]	Case report	47-year-old female with BED and severe depression	NAcc (bilateral)	Bipolar 3 V/90 µs/130 Hz	Weight loss (2.8 kg/month), affective improvement and decrease of binge eating behaviors

AN: anorexia nervosa; AN-R: anorexia nervosa restricting subtype; AN-BP: anorexia nervosa binge/purge subtype; BED: binge eating disorder; OCD: obsessive-compulsive disorder; MDD: major depressive disorder; GAD: generalized anxiety disorder; SUD: substance use disorder; PTSD: post traumatic stress disorder; BMI: body mass index; BNST: bed nucleus of the stria terminalis; SCC: subgenual cingulate cortex; NAcc: nucleus accumbens; MFB: medial forebrain bundle; ALIC: anterior limb of internal capsule; LHA: lateral hypothalamus; PET: positron emission tomography. * We retained the main stimulation parameter after adjustments: amplitude/pulse width/frequency. ^1,2^: Partial overlap on sample; ^3^: in a second phase, patients will be included in a randomized trial with two arms (ON/OFF or OFF/ON).

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
