# Peer review of "Brain Stimulation in Eating Disorders: State of the Art and Future Perspectives"

_jcm, 2020, doi:10.3390/jcm9082358_

Round 1
Reviewer 1 Report
In this perspective article about neuromodulation on eating disorders, the authors do a very nice job of summarizing the current state of research on these treatment modalities, which needs expansion. They also attempt to comment more on ways to improve these modalities for use in treatment, but this, though interesting, does dilute rather than improve the utility of their manuscript. It is important to define the goal for the article one is writing, and then stick to the format in order to ensure the reader can learn from what you write. If this is a systematic review that suffers from not having enough data sources to review, that is ok. If this is an opinion or perspective through which the authors want to promote the potential utility of neuromodulatory treatment for eating disorders, that is also ok, and it allows the authors to make more statements without an evidence base (as is done throughout the current manuscript). But it would be better to stick to one, and that is to the author’s discretion. As such, I recommend the following changes.
- Enhanced discussion of co-morbidities and how to disentangle them from eating disorders. Given that many of the treatments discussed are actually approved for use in depression, and as mentioned, a substantial number of people with ED also have co-morbid depression, it would be more relevant to add their considerable knowledge to this question, rather than re-summarize what is known about targeting location from other literature. This could be done as well for OCD. This would tie together the authors point about heterogeneity of illness with the thrust of the article about how to make use of these treatments for other purposes in a way that is not currently coming across.
- Consistent reporting of location and sidedness of treatment. This is inconsistent through the manuscript, but is relevant to report for each instance. I recommend also incorporating this discussion into each section, and therefore engaging the reader in the relevance of reporting it as the review-perspective flows, rather than having it be a standalone section in the discussion. As it is, the section on “sites of modulation” does not add much to the manuscript (though the figure is helpful, the captions are very hard to read). In addition, consistent reporting of age of treatment is relevant, since there are differences both in illness presentation and treatment response between adolescents and adults, as reported, and therefore cannot be lumped together. These points should also be added to table 1.
- The section on VNS in eating disorders seems unnecessary. Furthermore the inclusion of irrelevant research serves to confound the point of the manuscript as a whole. It is ok to restrict to writing about eating disorder treatment. It is important to consider what is different (as well as what is the same!) in an eating disorder vs what is seen in other disorders. This is relevant to point 1 above.
- The section on neuroanatomical targets for NIBS is missing a thoughtful interpretation of how these targets are interconnected. The authors make some comments about circuits in this section, but as it seems beyond the scope of evidence related to eating disorders, is not entirely useful to the overall manuscript as written. The section on neuroanatomical targets for DBS is generally not relevant to the manuscript as written as it is all conjecture as to how it would relate to eating disorders, and needs to be either explicitly connected, or removed.
- The perspective section is generally good, but again, must be connected carefully to the overall goal of the article. Because the section between the review component and this one is markedly tangential, it does not quite fit the flow of the article. Furthermore, the conclusion doesn’t accurately represent what was posited through the entirety of the manuscript, and is therefore inconsistent with the actual evidence.
- There are consistent small and large grammar and syntax errors throughout that are distracting and require attention.
Author Response
- Enhanced discussion of co-morbidities and how to disentangle them from eating disorders. Given that many of the treatments discussed are actually approved for use in depression, and as mentioned, a substantial number of people with ED also have co-morbid depression, it would be more relevant to add their considerable knowledge to this question, rather than re-summarize what is known about targeting location from other literature. This could be done as well for OCD. This would tie together the authors point about heterogeneity of illness with the thrust of the article about how to make use of these treatments for other purposes in a way that is not currently coming across.
Response: Discussion has been shortened in order to be more focused. Many sections have been deleted. Reviewer can find all modifications (both additions and deletions) in the manuscript.
We have also added a paragraph regarding comorbidities.
Psychiatric comorbidities are the norm in people with eating disorders (>70%) [1]. Binge disorders are often comorbid with a substance use disorders [1]. Binge eating is also frequently compared to addictions, based on the evidence that they share common characteristics, such as escalating frequency of the behavior, ambivalence towards treatments and frequent relapses [116,117]. Also, brain stimulation is a successful strategy in MDD. ECT is recommended in most guideline and rTMS in some of them [118]. Given the high rate of these comorbid disorders in patients with EDs, some of the published studies were built on this background. Protocols targeting DMPFC in BN were adapted from substance use disorders studies, ECT studies on severe MDD have been one of the main drivers to begin ECT in patients with MDD and AN. In DBS, AN comorbid with MDD and OCD was one of the reasons to target the SCC. In their study targeting two areas in the same trial for the first time, the targets were chosen based on two of the major psychiatric co-morbidities associated with AN: SCC for affective disorder (n=4) and NAcc for anxiety disorder (n=4) [99]. Indeed, most studies reported positive effects on depressive symptoms with rTMS [27], tDCS [56], ECT [62], and DBS [95]. Future studies in EDs might benefit from this knowledge in brain target selection (figure 1), in studies design (add on with another modalities of treatment, inclusion and exclusion criteria...), or in the stimulation parameters: number of sessions, type of coil or electrode, stimulation duration... Nevertheless, regarding the mood component, there is a crucial methodological problem, because of the simultaneous improvement of ED weight and mood features, a specific effect on ED cannot be identified. Systematic measurement of depressive symptoms associated with subgroup analyses of patients without depression will make it possible to address this problem.
- Consistent reporting of location and sidedness of treatment. This is inconsistent through the manuscript, but is relevant to report for each instance. I recommend also incorporating this discussion into each section, and therefore engaging the reader in the relevance of reporting it as the review-perspective flows, rather than having it be a standalone section in the discussion. As it is, the section on “sites of modulation” does not add much to the manuscript (though the figure is helpful, the captions are very hard to read). In addition, consistent reporting of age of treatment is relevant, since there are differences both in illness presentation and treatment response between adolescents and adults, as reported, and therefore cannot be lumped together. These points should also be added to table 1.
Response: we thank the reviewer for these suggestions. We have deleted the section "site of modulation" and incorporated some parts of it in each result’s sections. Reviewer can find all modifications (both additions and deletions) in the manuscript. Tables have also been incorporated reporting information regarding sample and treatment. To date, studies only focused on adults? This has been clarified line 428:
Finally, in most of the studies reviewed here, samples included only adults and mainly patients with severe, chronic ED.
Finally, the quality of figure 1 has been improved to make the caption more readable
- The section on VNS in eating disorders seems unnecessary. Furthermore the inclusion of irrelevant research serves to confound the point of the manuscript as a whole. It is ok to restrict to writing about eating disorder treatment. It is important to consider what is different (as well as what is the same!) in an eating disorder vs what is seen in other disorders. This is relevant to point 1 above.
Response: We have simplified this section according to the recommendations of the reviewer. In this section we have also added a few points about the modulation sites initially present in the discussion
To date, no study assessed VNS effects in patients with EDs [78]. However, a growing body of suggests the relevance of VNS in patients with ED. Some studies in animal models showed an association between VNS and reduction in food intake and/or weight loss, suggesting that vagal stimulation might mediate satiety signals (for review see McClelland 2013 [79]). Several fMRI studies have also shown that VNS modulates the activity in brain regions related to the processing of afferent vagal signals and interoception, such as thalamus, precentral gyrus and insular cortex [80-82]. A recent study demonstrated that transcutaneous VNS improves interoceptive accuracy [83].This is very valuable point given the central role of interoception in ED [84-86].
- The section on neuroanatomical targets for NIBS is missing a thoughtful interpretation of how these targets are interconnected. The authors make some comments about circuits in this section, but as it seems beyond the scope of evidence related to eating disorders, is not entirely useful to the overall manuscript as written. The section on neuroanatomical targets for DBS is generally not relevant to the manuscript as written as it is all conjecture as to how it would relate to eating disorders, and needs to be either explicitly connected, or removed.
Response: We have deleted section "site of modulation" and incorporated some parts of it in each result’s sections. The reviewer can find all modifications (both additions and deletions) in the manuscript. We have tried to keep these new paragraphs more focused.
- The perspective section is generally good, but again, must be connected carefully to the overall goal of the article. Because the section between the review component and this one is markedly tangential, it does not quite fit the flow of the article. Furthermore, the conclusion doesn’t accurately represent what was posited through the entirety of the manuscript, and is therefore inconsistent with the actual evidence.
Response: We have slightly modified the perspective section and part of the conclusion has been removed. The reviewer can find all modifications (both additions and deletions) in the manuscript.
- There are consistent small and large grammar and syntax errors throughout that are distracting and require attention.
Response: The manuscript has now been reviewed by a professional free-lance editor.
Reviewer 2 Report
This is a interesting and comprehensive review on the use of brain stimulation methods to treat eating disorders. It is clinically relevant and draws perspectives for future research.
I would propose to use the term brain stimulation instead of neuromodulation. Neuromodulation is too broad, e.g. stress or medication can work as neuromodulators. Moreover, this will increase the searchability of the paper.
The authors might consider to add a paragraph on depression as comorbidity in ED. Most interventions also target depression and this might be a crucial methodological problem. This problem is mentioned in many parts of the manuscript, however, a comprehensive discussion would be helpful.
I would propose a professional language editing.
Author Response
Reviewer 1
This is an interesting and comprehensive review on the use of brain stimulation methods to treat eating disorders. It is clinically relevant and draws perspectives for future research.
Response: We thank the reviewer for this comment.
I would propose to use the term brain stimulation instead of neuromodulation. Neuromodulation is too broad, e.g. stress or medication can work as neuromodulators. Moreover, this will increase the searchability of the paper.
Response: The term brain stimulation has been prioritized throughout the manuscript.
The authors might consider to add a paragraph on depression as comorbidity in ED. Most interventions also target depression and this might be a crucial methodological problem. This problem is mentioned in many parts of the manuscript, however, a comprehensive discussion would be helpful.
Response: Thanks for this relevant comment. A new chapter has been added in the Discussion section:
In addition, brain stimulation is a successful strategy for mood disorders and given the high rate of comorbid mood disorders in patients with EDs, many of the published studies built on this background. For instance, ECT studies on severe MDD have been one of the main drivers to begin ECT in patients with MDD and AN. In DBS, AN comorbid with MDD and OCD was one of the reasons to target the SCC. Indeed, most studies reported positive effects on depressive symptoms with rTMS [27], tDCS [41], ECT [54], and DBS [65]. This poses a crucial methodological problem because due to the simultaneous improvement of ED, weight and mood features, a specific effect on ED cannot be easily specified. Systematic measurement of depressive symptoms associated with subgroup analyses of patients without depression will make it possible to address this problem.
I would propose a professional language editing.
Response: The manuscript has been reviewed by a professional free-lance editor.
Reviewer 3 Report
This paper provides a comprehensive review of literature on neuromodulation in eating disorders. However, in my opinion, it can be greatly improved.
Firstly, it needs to be read and revised by a native English speaker.
Secondly, I think it could be shortened. Although it is a narrative review, there should be a table summarising the methods and findings from the papers that have been published/included in the review.
Thirdly, I accept that the field is in its relative infancy, however, I feel that the discussion could be more reflective; having said this I feel the perspectives section is very appropriate.
Fourth, to me, parts of the first section of the discussion feel somewhat repetitive of details that should be included in the results section/tables e.g., neuroanatomical targets.
Fifth, my sense is that some of the referenced literature particularly in the introduction e.g., background on eating disorders is rather dated and some additional and more current literature should be cited.
Author Response
Reviewer 2
This paper provides a comprehensive review of literature on neuromodulation in eating disorders. However, in my opinion, it can be greatly improved.
Response: We have tried to address the reviewer’s remarks. We hope the reviewer will find the manuscript improved.
Firstly, it needs to be read and revised by a native English speaker.
Response: The manuscript has been reviewed by a professional free-lance editor.
Secondly, I think it could be shortened. Although it is a narrative review, there should be a table summarising the methods and findings from the papers that have been published/included in the review.
Thirdly, I accept that the field is in its relative infancy, however, I feel that the discussion could be more reflective; having said this I feel the perspectives section is very appropriate.
Fourth, to me, parts of the first section of the discussion feel somewhat repetitive of details that should be included in the results section/tables e.g., neuroanatomical targets.
Response: We choose to address the second, third and fourth comment together because they are connected. The manuscript has been substantially modified. Four tables have now been included, some parts of the Results have been clarified, and the Discussion has been shortened. Given the high number of modifications, they are not reported here, but the reviewer can find all modifications (both additions and deletions) in the manuscript
Fifth, my sense is that some of the referenced literature particularly in the introduction e.g., background on eating disorders is rather dated and some additional and more current literature should be cited.
Response: We thank the reviewer for this comment. This has been fixed.
Round 2
Reviewer 3 Report
The authors have made improvements to the article. The addition of the tables has been very useful and greatly improves the results section. However, generally it has been difficult to review all of the amendments (in particular the additions) made by the authors due to the lack of tracked changes.
I have some specific minor comments remaining:
- I appreciate that a BMI below 18.5kg/m2 is suggested in the ICD-11 (but not in the DSM-5) for anorexia nervosa (line 36-37) - therefore, please include a reference to the diagnostic criteria used and add units of BMI. Also, I recommend that the authors change “becoming overweight” to “gaining weight” in line with diagnostic criteria.
- Comments on the tables:
- I think that citations should be included in the reference column of tables 1-4 so that the reader can identify the study in the bibliography.
- Table 1
- I noticed that there is a partial/full sample overlap between the following three studies McClelland et al. 2013, 2016, 2016 that has not been acknowledged.
- Given the authors comment in the discussion (lines 323-324), I think that the treatment characteristics column in rTMS table should mention whether treatment was MRI-guided.
- I also feel that it would be useful to add number of sessions to the treatment characteristics column.
- Table 2 - what order are the studies presented in? I wonder whether they need to be in date order as done in the other tables?
- Table 4 - I have noticed that the following study has not been included: Hayes et al. (2015). Brain Stimulation, 8(4):758-68. doi: 10.1016/j.brs.2015.03.005. which is an ancillary analysis of another study you have included.
- Table 5 - I feel that the details in this table are a little misleading, given that the authors are unclear on the overlap between studies (I appreciate that they have acknowledged this in the table footnotes). This is also particularly true given the variations in the study designs. As such, I wonder whether it would be useful for rTMS and tDCS rows to be divided into studies that assessed a single session and studies that used multiple sessions. My sense is that this is an important distinction, given that any feasible treatment with these modalities will likely be of multiple sessions (as the authors suggest in the discussion) and single session studies are more proof of concept tests.
- Results section
- I wonder if the sections of the results need to be reordered so that the non-invasive brain stimulation techniques follow on from one another and then the invasive ones to be in line with how you described the techniques in the introduction. Currently, the VNS section is in between the tDCS and ECT sections, which feels a little random.
- Line 278-279 - I think that it would be useful for the reader to add a citation to Park et al. here (reference number 71).
- Discussion Line 330 – I have noticed that this information is incorrect, several rTMS studies have used 20 sessions of rTMS e.g., McClelland et al. 2013 case report, McClelland et al. 2016 case series, Dalton et al. 2018/2020 RCT. Therefore, the comment in the discussion needs to be amended accordingly. Also, as above, I recommend that the authors review the studies in the rTMS and tDCS tables to ensure the correct number of sessions is reported and add this information to the treatment characteristics columns.
Author Response
1) I appreciate that a BMI below 18.5kg/m2 is suggested in the ICD-11 (but not in the DSM-5) for anorexia nervosa (line 36-37) - therefore, please include a reference to the diagnostic criteria used and add units of BMI. Also, I recommend that the authors change “becoming overweight” to “gaining weight” in line with diagnostic criteria.
Response: We have slightly modified manuscript to be consistent with the DSM and ICD definitions.
Anorexia nervosa (AN) is a multifactorial ED characterized by significantly low body weight for the individual’s height, age and developmental stage, intense fear of gaining weight despite obvious thinness, and extreme behaviors designed to lose weight
2) Comments on the tables:
- I think that citations should be included in the reference column of tables 1-4 so that the reader can identify the study in the bibliography.
Response: References has now been added in tables
- Table 1
- I noticed that there is a partial/full sample overlap between the following three studies McClelland et al. 2013, 2016, 2016 that has not been acknowledged.
Response: This is now acknowledged
- Given the authors comment in the discussion (lines 323-324), I think that the treatment characteristics column in rTMS table should mention whether treatment was MRI-guided.
Response: This is indeed an interesting point now added in table 1 (highlighted in yellow in manuscript)
- I also feel that it would be useful to add number of sessions to the treatment characteristics column.
Response: This information is now available in table 1 and 2 for all included studies (highlighted in yellow in manuscript)
- Table 2 - what order are the studies presented in? I wonder whether they need to be in date order as done in the other tables?
Response: we thank the reviewer for spotting this error. Table 2 is now presented by date order.
- Table 4 - I have noticed that the following study has not been included: Hayes et al. (2015). Brain Stimulation, 8(4):758-68. doi: 10.1016/j.brs.2015.03.005. which is an ancillary analysis of another study you have included.
Response: we thank the reviewer for bringing this study to our attention. Another study available on pubmed last week has also been added. (highlighted in yellow in manuscript)
- Table 5 - I feel that the details in this table are a little misleading, given that the authors are unclear on the overlap between studies (I appreciate that they have acknowledged this in the table footnotes). This is also particularly true given the variations in the study designs. As such, I wonder whether it would be useful for rTMS and tDCS rows to be divided into studies that assessed a single session and studies that used multiple sessions. My sense is that this is an important distinction, given that any feasible treatment with these modalities will likely be of multiple sessions (as the authors suggest in the discussion) and single session studies are more proof of concept tests.
Response: After discussion between co-authors, we feel, table 5 is superfluous with the addition of the new tables. We have deleted this table.
Results section
- I wonder if the sections of the results need to be reordered so that the non-invasive brain stimulation techniques follow on from one another and then the invasive ones to be in line with how you described the techniques in the introduction. Currently, the VNS section is in between the tDCS and ECT sections, which feels a little random.
Response: This has been done
- Line 278-279 - I think that it would be useful for the reader to add a citation to Park et al. here (reference number 71).
Response: This has been done.
- Discussion Line 330 – I have noticed that this information is incorrect, several rTMS studies have used 20 sessions of rTMS e.g., McClelland et al. 2013 case report, McClelland et al. 2016 case series, Dalton et al. 2018/2020 RCT. Therefore, the comment in the discussion needs to be amended accordingly. Also, as above, I recommend that the authors review the studies in the rTMS and tDCS tables to ensure the correct number of sessions is reported and add this information to the treatment characteristics columns.
Response: Numbers of sessions are now available in tables for all included studies (highlighted in yellow in manuscript) and the discussion has been amended.
With the exception of Dalton et al [29,44], the randomized controlled studies only proposed a limited number of pulses and 1 to 10 sessions (i.e. about 10,000 pulses maximum)